# A Fresh Perspective on Examining Population Emotional Well-Being Trends by Internet Search Engine: An Emerging Composite Anxiety and Depression Index

**DOI:** 10.3390/ijerph21020202

**Published:** 2024-02-09

**Authors:** Yu Wang, Heming Deng, Sunan Gao, Tongxu Li, Feifei Wang

**Affiliations:** 1Center for Applied Statistics, Renmin University of China, Beijing 100872, China; wangyu80@ruc.edu.cn; 2School of Statistics, Renmin University of China, Beijing 100872, China; dheming@ruc.edu.cn (H.D.); 2022103677@ruc.edu.cn (S.G.); tongxulee@ruc.edu.cn (T.L.)

**Keywords:** composite anxiety and depression index, group sentiment, search engine, natural language processing, emotion dictionary

## Abstract

Traditional assessments of anxiety and depression face challenges and difficulties when it comes to understanding trends in-group psychological characteristics. As people become more accustomed to expressing their opinions online, location-based online media and cutting-edge algorithms offer new opportunities to identify associations between group sentiment and economic- or healthcare-related variables. Our research provides a novel approach to analyzing emotional well-being trends in a population by focusing on retrieving online information. We used emotionally enriched texts on social media to build the Public Opinion Dictionary (POD). Then, combining POD with the word vector model and search trend, we developed the Composite Anxiety and Depression Index (CADI), which can reflect the mental health level of a region during a specific time period. We utilized the representative external data by CHARLS to validate the effectiveness of CADI, indicating that CADI can serve as a representative indicator of the prevalence of mental disorders. Regression and subgroup analysis are employed to further elucidate the association between public mental health (measured by CADI) with economic development and medical burden. The results of comprehensive regression analysis show that the Import–Export index (−16.272, *p* < 0.001) and average cost of patients (4.412, *p* < 0.001) were significantly negatively associated with the CADI, and the sub-models stratificated by GDP showed the same situation. Disposable income (−28.389, *p* < 0.001) became significant in the subgroup with lower GDP, while the rate of unemployment (2.399, *p* < 0.001) became significant in the higher subgroup. Our findings suggest that an unfavorable economic development or unbearable medical burden will increase the negative mental health of the public, which was consistent across both the full and subgroup models.

## 1. Introduction

China has experienced rapid economic development and tremendous social transformation in recent years. Such rapid social change can affect individual emotions, such as increasing insecurity, which is reflected in a marked increase in the incidence of mental illness [1,2]. A nationally representative study of Chinese residents found that 20.4% of people aged 18 and over suffered from anxiety, depression, or both [3]. Concurrently, anxiety is a common mental disorder ranked by the World Health Organization (WHO) as the ninth leading cause of health-related disability, accounting for 3.3% of the global disease burden [4]. Meanwhile, anxiety often co-exists with other mental disorders, especially depression [4]. The number of people with anxiety and depression disorders worldwide was about 246 million and 374 million by 2020, respectively [5].

The group-wide changes in anxiety and depression reflect the psychological characteristics of different regions, which provide a new dimension for the study of social situations [6]. Additionally, different cities have regional specificity, reflected in their development level, medical resources, Internet popularity, and other fields [2,7,8]. These urban characteristics, such as economic development and medical burden, are associated with negative public mental health [2]. Therefore, it is necessary to characterize group sentiment by city to make the analysis more nationally representative of the Chinese region [9].

Traditional assessments of anxiety and depression are facing challenges and difficulties when informing the trends of group psychological characteristics. These assessments are usually made using questionnaires or hospital tests, which require a large workforce and resources [10,11]. Also, there is widespread stigma associated with mental illnesses, such as anxiety and depression [12], significantly increasing the difficulty of detection and limiting its scope. The availability of large-scale mental health surveys in China is limited. The most representative survey, the China Mental Health Survey (CMHS), was conducted only in 2012–2015 [13,14]. Fortunately, with the popularization of the Internet and communication technology, more and more people are using location-based online media services (e.g., Twitter, Weibo) and search engines in their daily lives and posting emotionally-rich texts openly [15,16]. A growing body of research is crunching information from social media to extract specific and reliable emotions from people [17]. It has been confirmed that utilizing social media data from Twitter is effective in monitoring mental health [18]. In addition, Internet search trends are extensively employed in communication, medicine, health, business, and economic research. For example, using Google Search Trends to understanding mental health conditions or explore the correlations between COVID-19 vaccination choices and public mental health [19,20]. In another study, researchers [21] collected Google trends for five words, including “Depression”, “Insomnia”, “Autism”, “Psychologist” and “Psychiatrist” from 25 September 2016 to 19 September 2021 and investigated the future trajectory of public mental health. However, at present, there is no composite sentiment indicator that can reflect the anxiety and depression in China.

Analyzing unstructured emotionally enriched texts from location-based online media services or search engines is challenging. Fortunately, the rapid development of machine learning and text mining technologies provides a new platform for large-scale text mining and online data integration, helping to identify and analyze human behavior and group emotions [16,22]. Recent research proposed a model based on Skip-gram and CNN-BiLSTM for rapid analysis of Sina microblog emotion [23]. In addition, with the key dimensions of the Big Five model, including Openness to Experience, Conscientiousness, Extraversion, Agreeableness and Neuroticism, which are widely accepted as an adequate basis for the representation of human personality. In total, six different computational models were used to compare the ability to identify personality traits based on Facebook texts [24,25]. A system review has pointed out that natural language processing and machine learning have been widely applied in suicide research [26].

To develop a comprehensive indicator that effectively captures the anxiety and depression sentiments within the Chinese population and facilitates improved monitoring of changes in the psychological health levels across different provinces, this study introduced the Composite Anxiety and Depression Index (CADI). Utilizing the Baidu trend and a diverse range of emotionally rich texts, CADI can effectively reflect the periodic levels of public mental health in both temporal and regional dimensions. CADI can effectively reflect the periodic levels of public mental health in both temporal and regional dimensions. We verified the effectiveness of CADI using a representative data. Then, we used regression and subgroup analysis to explore the association between CADI, economic development, and medical burden. Our study offered a fresh perspective on examining population emotional well-being trends from the standpoint of online information retrieval, and we applied it for the first time to measure the mental health levels of populations in different regions of China.

## 2. Methodology

### 2.1. Construction of the Composite Anxiety and Depression Index

#### 2.1.1. Public Opinion Dictionary and Word Importance

The set, containing keywords highly correlated with the target words “Anxiety” and “Depression”, is referred to as the Public Opinion Dictionary (POD). To construct POD, the most critical issue was to obtain enough and accurate textual materials. We employed crawler technology to gather public social media data about anxiety and depression from three platforms. They are, respectively: (1) ***Weibo***, the largest social media platform in China, (2) ***Douban***, the largest community for book and movie sharing in China, and (3) ***Zhihu***, a high-quality question-and-answer community on the Chinese Internet. The time window for the crawler spans from 1 January 2020, to 30 June 2020. We filtered out text materials with fewer than 15 Chinese character counts. Then, we excluded text materials that was irrelevant to anxiety and depression. Finally, we obtained 7759 pieces of textual material, whose details were shown in (Appendix A).

Subsequently, based on the flow chart in Figure 1, we first utilized Jieba [27], one popular algorithm for Chinese natural language processing (NLP), to segment the Chinese text materials. We retained only nouns and adjectives with Chinese character counts greater than or equal to 2. We removed meaningless words and retained words with high frequencies. Then, we used the CBOW model proposed by [28], to calculate the importance of each word, whose details were shown in (Appendix A).

#### 2.1.2. Internet Search Trends

Different from the previous research that used Google trends, we focused on the Baidu trend. This is because most Chinese individuals utilize the Baidu search engine instead of Google. Baidu trends are a numerical tool, similar to Google trends, and can be used to reflect the intensity and interest of users searching for specific keywords. However, unlike Google trends, which only provide weekly data, Baidu trends can provide daily search data. In this work, we collected Baidu trends between 1 January 2016 and 31 December 2021 for each word in the POD. We received these data through the API interface provided by the Baidu Developer Platform.

#### 2.1.3. Calculation of the Composite Anxiety and Depression Index

Figure 2 depicts the pathway for constructing the Composite Anxiety and Depression Index. Let the correlation result computed by the CBOW model between the i-th word in the POD and its target words be α(i). Let β(i,y) be the yearly averaged Baidu trend for the i-th word in the y-th year. We then adjusted β(i,y) using the correlation of the i-th word as,
$$\varphi \left(i,y\right)=\alpha \left(i\right)\times \beta (i,y).$$

Next, the relative weight of the i-th word in the y-th year can be computed as follows:$$w\left(i,y\right)=\frac{\varphi (i,y)}{\sum_{i}\varphi (i,y)}.$$

The weight $w\left(i,y\right)$ can be explained as the proportion of the total impact of the i-th word on the public negative mental health in the y-th year.

Let $\gamma \left(r,i,y\right)$ be the yearly averaged Baidu trend of the i-th word in region r in the y-th year. Let $\;\delta \left(r,\;y\right)$ be the total population of region *r* in the y-th year. We then adjust $\gamma \left(r,i,y\right)$ by the corresponding population,
$$\bar{\gamma }\left(r,i,y\right)=\frac{\gamma \left(r,i,y\right)}{\delta \left(r,y\right)}.$$

Last, we compute the CADI for each region in each year. Let $CADI\left(r,y\right)$ be the CADI of region r in the y-th year. We use the efficacy coefficient linear weighting method [29] to calculate $CADI\left(r,y\right)$,
$$CADI\left(r,y\right)=\sum_{i}\left(\frac{\bar{\gamma }\left(r,i,y\right)-min\left(\bar{\gamma }\left(r,i,y\right)\right)}{max\left(\bar{\gamma }\left(r,i,y\right)\right)-min\left(\bar{\gamma }\left(r,i,y\right)\right)}\times \tau +100-\tau \right)\times w\left(i,y\right).$$

The larger the CADI, the more serious the negative mental health of the public. In the above equation, $min\left(\bar{\gamma }\left(r,i,y\right)\right)$ and $max\left(\bar{\gamma }\left(r,i,y\right)\right)$ represent the minimum and maximum values of the yearly averaged Baidu trend of the i-th word in all regions in y-th year. The parameter τ is used to control the range of CADI. In this work, we set τ = 40 to make CADI vary between 60 and 100.

To ensure vertical comparability of CADI, it is necessary to eliminate the impact of the growth in the number of Internet users on public opinion. To achieve this, we employ the growth rate of Internet penetration in China to adjust CADI. Let π(y) be the Internet penetration rate in China in the y-th year. Then, we compute the adjusted CADI as:$$\tilde{CADI}\left(r,y\right)=CADI(r,y)\times \frac{\pi (y)}{\pi (y-1)}$$

### 2.2. Explanatory Variables for the Composite Anxiety and Depression Index

Explanatory variables within economic development and medical burden from 2016 to 2021 were obtained from the Chinese provincial statistical yearbooks or the Chinese health statistical yearbooks. We then collected data for 31 prefecture-level provinces or municipalities from 2016 to 2021. The total sample size was 186.

In addition, the rate of unemployment is a well-studied factor [30,31,32]. We then incorporated the rate of unemployment as an explanatory variable. Descriptive statistics and the meaning of these explanatory variables are provided in the Appendix A. Detailed descriptive statistics of all explanatory variables are provided in Appendix A, while the definitions and meanings of these explanatory variables can be found in Appendix A.

### 2.3. Modeling Strategy

We first employed the linear regression model to investigate the relationship between explanatory variables and CADI. Linear regression is a well-established technique with strong explanatory power, making it widely used in practice. Ref. [33] utilized linear regression and generalized additive models to evaluate the link between 25(OH)D3, 25(OH)D2, and depression. Ref. [34] employed linear mixed models and linear regression models to explore the effects of prenatal maternal depression, anxiety, stress, and postpartum depression on early neurodevelopment and gender dimorphism in infants. To account for the temporal factor, we introduced a dummy variable for each specific year. Moreover, we utilized the quantile regression model to enhance the reliability and interpretability of our research results. Quantile regression is not affected by outliers and thus results in more robust results.

## 3. Results

### 3.1. Correlation between CADI and Prevalence of Mental Disorders

We further explored the potential association between the CADI and the prevalence of mental disorders across different regions. The China Health and Retirement Longitudinal Study (CHARLS) stands out as one of the few sources providing high-quality microdata representative of Chinese households and individuals [35]. In this study, we utilized data relevant to the prevalence of mental disorders from the 2020 survey conducted by CHARLS, which was released in 2023.

Utilizing sample weights provided by CHARLS, we calculated the weighted prevalence of mental disorders for each province. Figure 3 showed the weighted prevalence of mental disorders and the 2020 CADI, *** represented the *p*-value less than or equal to 0.001. Notably, a clear linear trend is observable, indicating a positive association between CADI and the prevalence of mental disorders. Regions with higher CADI values exhibit higher rates of mental disorders. To quantify this trend, we computed Pearson, Kendall, and Spearman correlation coefficients, yielding values of 0.5841, 0.5498, and 0.7478, respectively, all significant at the 0.05 level. These statistical findings provide evidence supporting the effectiveness of CADI, suggesting its capacity to reflect variations in the prevalence of mental disorders across different regions.

### 3.2. Spatiotemporal Trends of CADI

Figure 4 showed the contrast of CADI across different years and regions. We found that, from 2016 to 2021, China experienced fluctuations in negative mental health, but an overall upward trend can be observed. It was worth noting that the Northeast region, particularly Beijing and Tianjin, exhibited relatively serious negative mental health status among the population, while the Southwest region, exemplified by Tibet and Qinghai, exhibited relatively high levels of negative mental health. Additionally, economically developed areas, such as Shanghai, displayed the highest levels of negative mental health among eastern coastal regions. Starting from 2016, the level of negative mental health in the eastern coastal areas had increased to some extent, and after a brief decline in 2020, it had rapidly risen again. Negative mental health remained relatively stable in the central region from 2016 to 2019 but increased rapidly after 2020. Notably, the underdeveloped areas in the western region displayed a higher level of negative mental health among the population than at the national level.

### 3.3. Regression Model Results

We then explored the association between CADI, economic development, and medical burden. Table 1 provided details on the models. We fitted linear regression models (LRM) for an overall model (Overall) that incorporated all provinces. The Overall model results revealed that the Import–Export Index (IE) had a significant negative impact on negative mental health (*p* < 0.0001). Specifically, for each unit increase in the IE level, CADI decreased by 16.272. Additionally, there was a significant positive correlation between the Average Cost of Patients (ACP) and negative mental health (*p* < 0.0001). For every unit increase in ACP level, CADI increased by 4.412. These findings suggested that CADI was associated with the level of international trade in a region and the direct cost of treating diseases.

To minimize the impact of socioeconomic status and related factors [36,37], we divided all provinces into two subgroups: a low per capita GDP group (subgroup1) and a high per capita GDP group (subgroup2) based on the averaged six-year national per capita GDP, see Appendix A for more information. We then fitted separate LRM for each group. For the subgroup 1, we found that Disposable Income (DI) and IE had significant negative impacts on negative mental health, whereas ACP had a significant positive effect. In comparison to the Overall results, the impact of IE on CADI had weakened. In contrast, the effect of DI was significant in the subgroup1. Additionally, the degree of influence of ACP on negative mental health increased to a certain extent. For each unit increase in ACP, CADI increased by 8.210.

For the subgroup2, the impact of ACP on negative mental health remained significant. Meanwhile, the Rate of Unemployment (RU) also significantly positively affected negative mental health. For every unit increase in RU level, CADI would increase by 2.399, and the significance level was less than 0.01. To enhance the robustness of our results, we used a quantile regression model (QRM, quantile point is 0.50) for analysis. The quantile and linear regression models showed consistent results.

## 4. Discussion

The CADI can reflect public mental health at temporal and geographical dimensions. At the temporal level, the CADI indicated an overall increasing trend in negative mental health levels across various regions of China from 2016 to 2021. At the geographical level, the CADI demonstrated regional disparities in public mental health, with more prominent negative mental health observed in the northwest border areas and the southeast coastal regions. A representative cross-sectional epidemiological study of mental health in China revealed a continuous rise in public psychological stress levels over the past several decades [13]. It was noticed that a more severe psychological state in border regions [38]. Additionally, higher rates of mental disorders were observed among both adults and children from prosperous coastal areas [39]. The characteristics of public mental health obtained from the CADI are consistent with previous empirical studies. In addition, there is a significant positive correlation between CADI and the prevalence of mental illnesses. Therefore, it is valid to employ the CADI as a measure of public negative mental health.

We found higher levels of negative mental health in both relatively developed areas (e.g., the eastern coastal areas) and less developed areas (e.g., southwest areas). At the national level, the CADI showed a U-shaped distribution with the GDP, and our findings were consistent with the previous literature. An analysis of more than 450,000 samples found that low income was associated with lower life evaluations and lower emotional well-being, which increased as income increased. However, after reaching a certain income level in the current year ($75,000 in the US), life happiness did not increase further, and the relationship between the two showed a strong convex function [40].

Using GDP stratification can reduce the influence of socioeconomic status and its accompanying factors (e.g., health disparities and living standards) on the association results [36,37]. ACP showed a significant negative correlation with the CADI in both the full and hierarchical models of linear and quantile regression (*p* < 0.05). This suggests that negative mental health is consistently associated with medical spending in past stages of development, regardless of the characteristics of economic growth in different regions. This aligns with the expectations that China has faced the COVID-19 pandemic and increasingly severe zoonosis (e.g., H5N1, H7N9) over the past few years. This series of diseases with apparent risks brought the population great emotional and economic burdens [41,42]. On the other hand, with the increased availability of CMIs, people receive more medical financial protection. Still, the actual increase in expenditure and economic burden may be due to patients being treated more frequently in higher-grade hospitals [43].

In addition, compared with the average level, the change of negative sentiment of groups in low GDP areas is more sensitive to the shift in Disposable Income and the Average Cost of Patients. This suggests that poorer areas are more concerned about income levels and healthcare burdens. The lowest income groups have been disproportionately affected by economic expansion. Economic expansion has brought more resources to high-income people than those at the bottom of the income scale, allowing them to adapt to macroeconomic changes [44]. Catastrophic health spending is also concentrated among poorer households [43]. On the other hand, in regions with higher GDP, the unemployment rate shows a significant positive correlation with CADI, which is consistent with previous studies [45]. Unemployment is a stressful life event, involving financial stress and potential benefit deprivation, which is more pronounced in urban areas with higher standards of living and can be accompanied by higher rates of suicide [45,46].

It is noteworthy that the CADI differs significantly from previous studies that solely relied on search engine trends to represent the public mental health of a particular region. Although previous studies have begun to use Internet platforms to construct representative search indicators through the thesaurus and Baidu trend and to evaluate the anxiety situation of each city further [2]. Our Public Opinion Dictionary (POD) was obtained based on the major media of Chinese public opinion. Compared with the mode of specifying the word list in the past, this method guarantees the habits of the times and the characteristics of the population.

Further, we used word correlation and frequency to consider the weight of search terms against target words (“Anxiety” and “Depression”), which reduced the interference of marginal and low-frequency words. On the other hand, with the rapid development of China, the Internet penetration rate of the population in different regions was constantly changing vastly, which was also covered by our CADI model [36]. It is necessary because there may be a correlation between the population size (or the popularity of the Internet) and changes in regional economic levels, search frequency, and even subjective happiness [36,37,38]. Therefore, our CADI has evolved from past methods to be more adaptable and representative.

Although a large sample population and a mature indicator construction process improve the application value of correlation results, our approach still has some limitations. Firstly, our information collection platform covers a relatively limited range of people, which could affect the representativeness and critical stability of the samples concerning age range and whether they are Internet users [47]. Secondly, using the Baidu trends may be affected by “Baidu search suggestions”, a common problem in search-based composite indexes. It’s worth noting that searches related to anxiety and depression may not always be directly associated with individuals experiencing anxiety or depression. Future research should explore the application of CADI to the longitudinal causal framework or predictive analysis. The calculation of CADI across different regions and over a more extended period also requires further validation. However, with the continuous improvement of the network information platform and computer computing power, our Composite Anxiety and Depression Index, construction process, and correlation results will provide extensive reference value.

## 5. Conclusions

In summary, this study developed the Public Opinion Dictionary (POD) and the Composite Anxiety-Depression Index (CADI) using social network data for the first time. POD provides a comprehensive dictionary that captures population anxiety and depression, and CADI can effectively reflect the periodic levels of public mental health in both temporal and regional dimensions. We then investigated the association between CADI and economic and medical indicators in China. Our findings suggest that CADI may be linked to both the regional trade level and the direct cost of disease treatment, which was consistent across both the full and subgroup models. While our research had certain limitations, our analytical procedures and findings provided valuable insights into the prevalent negative emotions in the population. These findings could serve as a reference for analyzing economic trends and formulating relevant policies.

## Figures and Tables

**Figure 1 ijerph-21-00202-f001:**
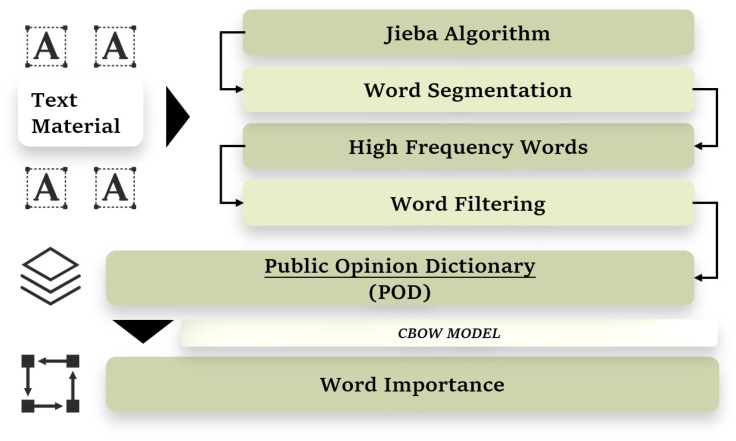
The Workflow of Constructing the Public Opinion Dictionary.

**Figure 2 ijerph-21-00202-f002:**
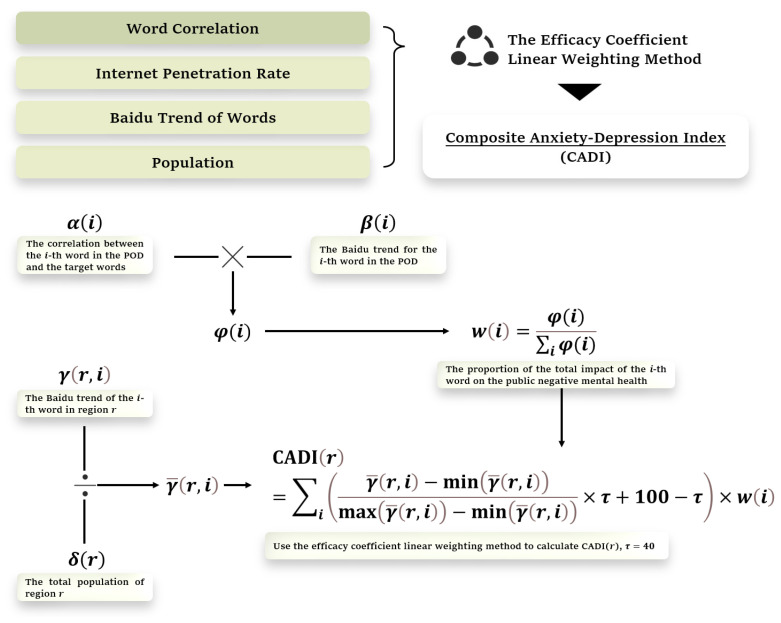
The Pathway for Synthesizing the Composite Anxiety and Depression Index.

**Figure 3 ijerph-21-00202-f003:**
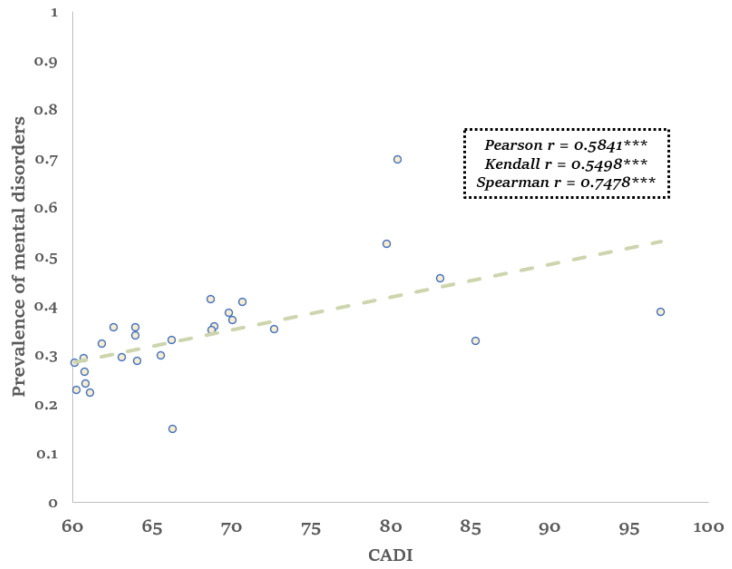
Linear Trend Between CADI and Prevalence of Mental Illness. Note: *** represented the *p*-value less than or equal to 0.001.

**Figure 4 ijerph-21-00202-f004:**
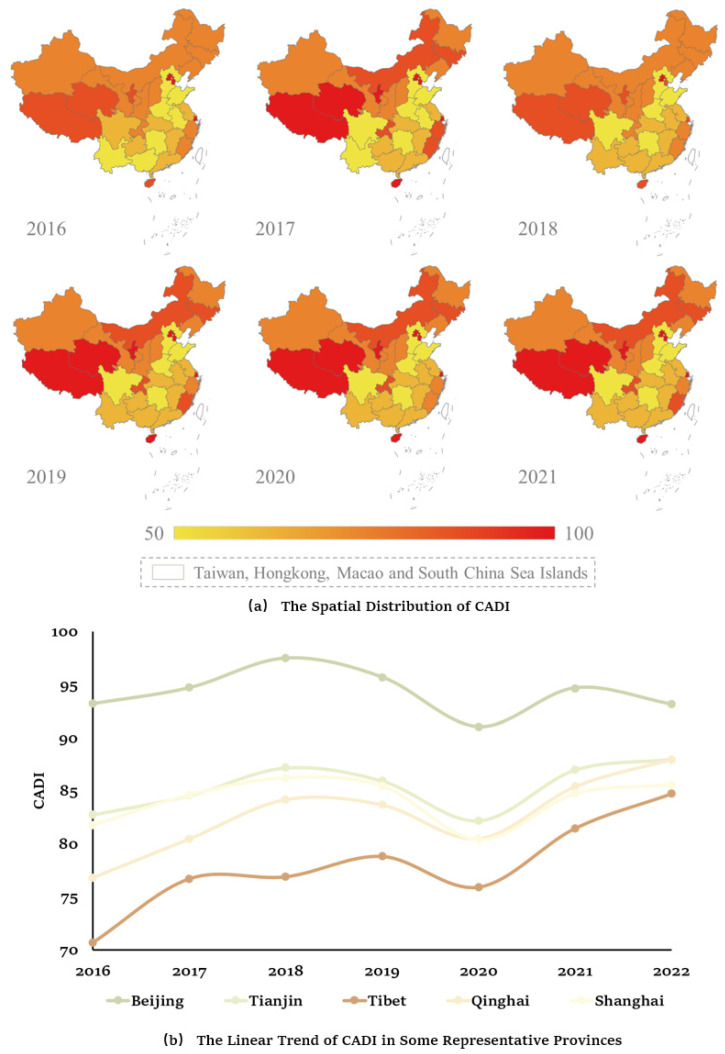
The Spatial Distribution and Linear Trend of CADI.

**Table 1 ijerph-21-00202-t001:** Results of Regression Model.

	LRM			QRM		
Group	Overall	Subgroup1	Subgroup2	Overall	Subgroup1	Subgroup2
Economic Development						
DI	−1.587	−28.389 ***	1.588	1.066	−23.367 **	1.366
CPI	0.142	0.388	−0.073	0.201	0.634	−0.007
CGR	−2.568	−23.537	−7.069	−14.783	−37.098	−15.386
IE	−16.272 ***	−9.167 **	13.329	−13.490 ***	−11.600 **	12.852
Medical Burden						
ACP	4.412 ***	8.210 ***	3.965 ***	3.697 ***	6.132 ***	3.951 ***
RHE	−0.005	0.100	−0.215	−0.951	−0.982	−1.218
RD	21.961	33.769	120.374	112.412	66.792	189.614
RU	−0.697	−1.782	2.399 **	−0.84	1.934	3.867 *
year2	1.102	−12.074 **	12.836 ***	5.105	−13.183 *	13.704
year3	6.902 **	−4.011	10.477 ***	9.504 **	−4.434	11.523
year4	6.383 **	−1.592	9.692 ***	7.879 **	−3.466	12.109 *
year5	2.947	−0.056	8.974 ***	4.616	0.268	11.361
_cons	49.176 *	84.993 *	12.51	43.749	67.618	9.737
R-squared	0.674	0.577	0.871	-	-	-

Note. * *p* < 0.05, ** *p* < 0.01, *** *p* < 0.001.

## Data Availability

The dataset used in this study came from public sources, such as statistical yearbooks and the Internet. We did not have any special access privileges that others would not have.

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
