# Peer review of "A Fresh Perspective on Examining Population Emotional Well-Being Trends by Internet Search Engine: An Emerging Composite Anxiety and Depression Index"

_ijerph, 2024, doi:10.3390/ijerph21020202_

Round 1

Reviewer 1 Report

Comments and Suggestions for Authors

Dear authors, 

Please find the revision of your article, 

Sincerely

A Fresh Perspective on Examining Population Emotional Well-being Trends by Internet Search Engine: An Emerging Composite Anxiety and Depression index

Review for Mental Health

The summary seems incomplete. What is the hypothesis, what are the main results?

The first paragraph of the introduction would benefit from being reorganized. It seems useful to start the introduction with the contextual elements, namely the fact that China has experienced rapid acceleration and profound changes, which has had effects on mental health, then to focus secondly on anxiety in particular.

The authors quickly make a connection between biases linked to anxiety assessment tools and the fact that people use location-based applications whose content can be emotionally charged. The connection is not obvious. The anxiety assessment is standardized and for diagnostic purposes. It cannot be compared directly with the writings of people on the Internet... Authors should be more careful and avoid this shortcut. They should also support their statement so that we understand the link they are establishing.

The authors mention interesting correlational studies which highlight a potential link between the terms used for Google search and economic indicators. In what direction does this correlation go? Can the authors elaborate so that we understand why these studies are an important point of support for theirs?

The authors mention the Big Five model without developing it. It would be necessary to explain what it is and add bibliographic references for readers.

The problem is not clear. What is the theoretical controversy in which this study is part? What is the general hypothesis and what are the operational hypotheses?

In the methodology, the authors should add a section relating to the sample (recruitment, sampling, ethical and professional precautions, inclusion and exclusion criteria, etc.)

To create the dictionary, what filters were applied? (Date of creation of the writing for example? We can imagine that the terminology is more or less intense depending on the temporal context (Covid…). 

In the section on Internet search trends, the authors inject important theoretical information. It would be appropriate to place them in the introduction and to articulate them with the rest in order to highlight the problem.

Why did you choose this period 2016 to 2021? Could the authors justify this choice?

The discussion is rich but the reader has difficulty finding their way around it. If the authors had posed the different operational hypotheses and analyzed them one by one, the results of this study would be clearer. 

The article should be brought up to the bibliographic standards of the journal.

Reviewer 2 Report

Comments and Suggestions for Authors

Thank you for the opportunity to review this paper. The manuscript presents data from the CADI and corresponding depression and anxiety in China. Overall, I really liked the idea of the paper. I am not an expert in machine learning nor the statistics used to create the CADI, and, hopefully, the editor can find another who is proficient in those methods. However, I had some questions that need addressed in the current study. 

1. One major issue that I had with the paper is using sociological data to make conclusions about individuals. The CADI is a sociological indicator of searches related to depression and anxiety. Moreover, the paper addresses important macro-level variables, such as GDP. I have dealt with sociological data like this before and there is so many alternatives and other variables to address that making such connections is tough. For instance, the authors admit that the CHARLS assesses micro-level data regarding depression and anxiety; however, as I already indicated, the predictors are macro-level. This creates a situation where the ecological fallacy is paramount to consider. A better approach would have been to find another source of depression/anxiety prevalence that is assessed using sociological methods as the outcome. 

2. The second issue that I had was that the authors are equating search engine data as an indicator of depression or anxiety. There are lots of reasons to use search engines regarding depression and anxiety that do have little to do with having depression or anxiety. For instance, before the first case of COVID-19 was in the US, I am sure there were a lot of Internet searches for COVID-19 due to concern over that disease. Related to depression, what is a student is writing a report on depression and starts on a search engine? What if a pet owner uses a search engine to search for dog anxiety? In short, it is a big stretch to assume that searching for a key word is indicative of having a potential trait or symptom indicated by that search. 

3. The introduction notes the limitations in measuring depression and anxiety via self-report, practitioner report, etc.; however, what is left? Even if we come to an agreement that the CADI is a valid measure of depression or anxiety, then how is this any better than those methods? I would argue that with privacy issues in some countries (I do not know about China), then the CADI would be difficult to ascertain - rendering this measure as potentially more problematic. Moreover, if a mental health counselor sees that a child scores 6 on the CADI, it is not likely that they will know how to interpret and treat that number. In short, the decades of quality work with more traditional measures, metrics, and methods are not invalidated.

4. Finally, the authors put a large focus on China. I understand that the reason for this is that the data collected was from Chinese participants and searches, and the program was made using algorithms based on search engines primarily used in China; however, for this paper to be meaningful to the scientific community, more discussion is needed as to whether these results would generalize beyond China. Would a very similar algorithm and procedure work if google was used on a US sample, for instance?

Round 2

Reviewer 1 Report

Comments and Suggestions for Authors

Thank you